# Carbapenem Use in the Last Days of Life: A Nationwide Korean Study

**DOI:** 10.3390/antibiotics12060964

**Published:** 2023-05-26

**Authors:** Yu Mi Wi, Ki Tae Kwon, Cheon-Hoo Jeon, Si-Ho Kim, Soyoon Hwang, Sohyun Bae, Yoonjung Kim, Hyun-Ha Chang, Shin-Woo Kim, Hae Suk Cheong, Shinwon Lee, Dong Sik Jung, Kyung Mok Sohn, Chisook Moon, Sang Taek Heo, Bongyoung Kim, Mi Suk Lee, Jian Hur, Jieun Kim, Young Kyung Yoon

**Affiliations:** 1Department of Internal Medicine, Samsung Changwon Hospital, Sungkyunkwan University School of Medicine, Changwon 51353, Republic of Korea; yumi.wi@samsung.com (Y.M.W.); cheonhoo144.jun@samsung.com (C.-H.J.); siho.kim@samsung.com (S.-H.K.); 2Division of Infectious Diseases, Department of Internal Medicine, Kyungpook National University Chilgok Hospital, School of Medicine, Kyungpook National University, Daegu 41404, Republic of Korea; sy13111@knu.ac.kr; 3Division of Infectious Diseases, Department of Internal Medicine, Kyungpook National University Hospital, School of Medicine, Kyungpook National University, Daegu 41944, Republic of Korea; meditwo@knu.ac.kr (S.B.); kimyj@knu.ac.kr (Y.K.); changhha@knu.ac.kr (H.-H.C.); ksw2kms@knu.ac.kr (S.-W.K.); 4Division of Infectious Diseases, Department of Internal Medicine, Kangbuk Samsung Hospital, Sungkyunkwan University School of Medicine, Seoul 03181, Republic of Korea; philliper@naver.com; 5Department of Internal Medicine, Pusan National University School of Medicine and Medical Research Institute, Pusan National University Hospital, Busan 50612, Republic of Korea; ebenezere@pusan.ac.kr; 6Department of Internal Medicine, Dong-A University College of Medicine, Busan 49201, Republic of Korea; dsjung@dau.ac.kr; 7Department of Internal Medicine, School of Medicine, Chungnam National University, Daejeon 35015, Republic of Korea; medone@cnuh.co.kr; 8Division of Infectious Diseases, Department of Internal Medicine, Inje University College of Medicine, Busan 47392, Republic of Korea; csmoon@paik.ac.kr; 9Division of Infectious Diseases, Department of Internal Medicine, College of Medicine, Jeju National University, Jeju 63241, Republic of Korea; cadevar@jejunu.ac.kr; 10Department of Internal Medicine, Hanyang University College of Medicine, Seoul 04763, Republic of Korea; sobakas@hanyang.ac.kr (B.K.); quidam76@hanyang.ac.kr (J.K.); 11Division of Infectious Diseases, Department of Internal Medicine, Kyung Hee University Hospital, Kyung Hee University College of Medicine, Seoul 02447, Republic of Korea; mslee7@khu.ac.kr; 12Department of Internal Medicine, Yeungnam University Medical Center, Daegu 42415, Republic of Korea; sarang7529@yu.ac.kr; 13Division of Infectious Diseases, Department of Internal Medicine, Korea University College of Medicine, Seoul 02841, Republic of Korea; young7912@korea.ac.kr

**Keywords:** handshake, carbapenem, antimicrobial stewardship programs

## Abstract

The appropriate use of carbapenem is a critical concern for patient safety and public health, and is a national priority. We investigated the nationwide status of carbapenem prescription in patients within their last 14 days of life to guide judicious-use protocols from the previous study comprised of 1350 decedents. Carbapenem use was universally controlled through computerised authorisation system at all centres during the study period. Carbapenem prescribing patterns and their optimality were evaluated. A total of 1201 patients received antimicrobial agents within the last two weeks of their lives, of whom 533 (44.4%) received at least one carbapenem. The median carbapenem treatment duration was seven days. Of the 533 patients receiving carbapenems, 510 (95.7%) patients had microbiological samples drawn and 196 (36.8%) yielded carbapenem-resistant pathogens. A total of 200 (37.5%) patients were referred to infectious disease (ID) specialists. Of the 333 patients (62.5%) who did not have ID consultations, 194 (58.2%) were assessed as “not optimal”, 79 (23.7%) required escalation, 100 (30.0%) required de-escalation, and 15 (4.5%) were discontinued. Notwithstanding the existing antibiotic restriction program system, carbapenems are commonly prescribed to patients in their last days of life.

## 1. Introduction

Carbapenems effectively treat serious infections because of their broad spectrum, covering multidrug-resistant (MDR) Gram-negative bacteria, such as extended-spectrum β-lactamase (ESBL)-producing or AmpC-hyperproducer Enterobacteriaceae, and nosocomial non-fermenters, such as *Pseudomonas aeruginosa* and *Acinetobacter baumannii* [1]. Over the past decade, the prevalence of MDR Gram-negative pathogens has considerably risen, contributing to the global escalation in carbapenem usage [2,3]. Carbapenems are the third most used antibiotic for community-acquired infections in the intensive care unit (ICU) (10.7%) and the first for hospital-acquired infections (21.5%) [3]. Excessive carbapenem use leads to an increased cost burden, adverse effects, and patient mortality [4]. Excessive carbapenem consumption is an important predisposing factor for worsening infection rates caused by multidrug-resistant *P*. *aeruginosa* (MRPA), *A*. *baumannii* (MRAB), and carbapenem-resistant Enterobacteriaceae (CRE) [5,6,7]. Recently, the growing incidence of carbapenem-resistant Gram-negative bacilli has become an urgent global healthcare challenge [8]. Therefore, the appropriate use of carbapenems is an important patient safety and public health issue, and a national priority.

Approximately 66% of patients in a French survey were prescribed carbapenems on an empirical basis for a median duration of eight days in the ICU [9]. In a previous study, we reported that carbapenems are prescribed to approximately half the patients (44.4%) in the highest amount (301.2 days of therapy (DOT) per 1000 patient-days) in the last two weeks of life [10]. A better understanding of carbapenem prescribing habits will help develop comprehensive recommendations for carbapenem use in patients in their last days of life.

We conducted a post-hoc analysis of a previous nationwide study to investigate the current carbapenem prescribing status to patients in their last days of life to guide the judicious use of carbapenems.

## 2. Results

### 2.1. Characteristics of Carbapenem Use in Patients in the Last Two Weeks of Their Life

A total of 1350 patients died at 14 hospitals during the study period. A total of 1201 patients received an antimicrobial agent during the last two weeks of their lives, of whom 533 (44.4%) received at least one carbapenem (Figure 1). The median carbapenem treatment duration was 7 (3–12) days. The most prescribed carbapenem was meropenem (n = 444, 83.3%).

The median patient age was 71.0 years. A total of 224 (42.0%) patients had an underlying cancer. Nearly two-thirds (n = 341, 64.0%) of the patients died due to an infectious disease. Cancer was the second most common cause of death (n = 110, 20.6%). At the time of death, 407 patients (76.0%) had LST documents. A total of 510 (95.7%) patients had microbiological samples drawn and 196 (36.8%) yielded carbapenem-resistant pathogens. Characteristics of patients receiving carbapenems in the 14 days before death were compared with those receiving non-carbapenem treatment. Variables with a *p* value < 0.05 in the bivariate analysis were included in the subsequent multivariate analysis. A logistic regression model revealed that infectious diseases (odds ratio (OR) = 4.16, 95% confidence interval (CI) 2.70–6.41; *p* < 0.001)) and isolated MDR pathogens (OR = 2.31, 95% CI 1.72–3.12; *p* < 0.001) were independent predictors of carbapenem use in patients in the 14 days before death. The goodness of fit of the final logistic regression model appeared to be satisfactory (Hosmer–Lemeshow statistic, χ2 = 8.766, *p* = 0.270). Of the 533 patients receiving carbapenems, only 200 (37.5%) were referred to ID specialists; formal ID consultations were not requested for 333 (62.5%) patients. Among these 333 patients, 79 (23.7%) were assessed as requiring escalation, 100 (30.3) as de-escalation, 116 (34.8%) as continuation, 15 (4.5%) as discontinuation, and 23 (6.9%) were not assessable (Table 1).

### 2.2. Comparison of Characteristics between “Optimal” and “Not Optimal” Carbapenem Prescriptions in Patients without an ID Specialist Consultation

After an ID specialist review, 194 (62.6%) prescriptions were assessed as “not optimal”, and 116 (37.4%) were assessed as “optimal”. Table 2 shows the characteristics of patients prescribed either “not optimal” or “optimal” antibiotics. The carbapenem treatment duration, sex, age, underlying comorbidities, cause of death, LST form completion, MDR pathogens isolated, and antibiotic class used showed no differences between the two groups. The number of antibiotic changes during the last 14 days of their life was significantly less in the “not optimal” group (odds ratio (OR) 0.83, 95% confidence interval (CI): 0.71–0.97, *p* = 0.023) than in the “optimal” group.

## 3. Discussion

This nationwide cohort study provided insights into the current practice of carbapenem use within the last few days of life. We demonstrated that carbapenems were frequently administered to patients within the last few days of life, and infectious diseases and isolated MDR pathogens are independent predictors of carbapenem use in the 14 days before death. Although carbapenem prescription was universally restricted by a computerised antibiotic control program, a considerable amount (62.5%) was prescribed without an ID specialist consultation, of which only a small portion of the carbapenem use (34.8%) was “optimal”.

Carbapenems are the drugs of choice for infections caused by MDR bacteria, such as ESBL-producers and several non-fermenters. However, studies have shown that there is a link between carbapenem use and resistance at both the individual and unit levels of infecting flora and gut microbiota [11,12,13,14]. In addition, carbapenem use has been associated with an increased risk of colonization of carbapenem-resistant *K. pneumoniae* in the ICU [15]. Our study showed that 533 (39.5%) patients received at least one carbapenem among the 1350 decedents during the last two weeks of their lives; microbiologic analysis of these patients yielded carbapenem-resistant pathogens in 36.8%. However, we did not investigate the causal relationship between carbapenem use and resistance. Numerous uncontrolled factors, such as infection control and the impact of non-carbapenem antibiotics, could have affected the emergence of carbapenem-resistant organisms [12].

Carbapenem use was controlled using a computerised antibiotic restriction program at all centres during the study period. Notwithstanding the existing antibiotic restriction program system and the high microbiological study performance (95.7%), only 116 (34.8%) prescriptions were assessed as “optimal” among the 333 carbapenem prescriptions issued without a formal ID specialist consultation. At most centres, a carbapenem prescription was issued by the attending physician without disruption. After this, ID specialists provided recommendations on whether carbapenem should be continued for a specific period or discontinued, using a computer-generated alert system and systemic post-prescription chart review within 72 h. In our study, 194 (58.2%) of the 333 patients without ID consults were assessed as “not optimal”: 79 (23.7%) required escalation, 100 (30.0%) required de-escalation, and 15 (4.5%) were discontinued. In addition, the number of antibiotic changes was significantly less in the “not optimal” group. Therefore, the study suggested that a computerised antibiotic restriction program did not work effectively on carbapenem prescribing to patients in their last days of life, and many physicians continued administering carbapenems although a change of antibiotic was needed. One study showed that approximately 30% of physicians intended to continue antimicrobial administration for patients at the end of therapy even after this was deemed medically futile [16]. In a previous nationwide cross-sectional survey, carbapenems were frequently chosen on an empirical basis for treating suspected infection. However, de-escalation is frequently not implemented despite isolates being susceptible to other drugs and the duration of carbapenem use was similar among patients with and without microbiological results [9]. Hence, the authors suggested that the antibiotic stewardship program (ASP) of carbapenem prescriptions may not be efficient in controlling its prescription, and antibiotic consultations may help in achieving de-escalation [9]. A previous study showed that antimicrobial use is influenced not only by ASP but also ID specialist consultations [6]. ID consultations lead to the reduction of carbapenem use, resulting in favourable outcomes such as shorter hospital stays and reduced mortality [6]. A study conducted in Germany demonstrated that prospective audits and feedback from ID specialists lead to the reduction of both the overall use of antimicrobial agents and the proportion of broad-spectrum antibacterial use [17]. A Swedish study established that an ID specialist-guided antimicrobial stewardship program consisting of prospective audits performed twice weekly profoundly reduced antibiotic use with no negative effect on patient outcomes [18]. ID-guided consultations for *Staphylococcus aureus* bacteraemia demonstrated better survival benefits and improved use of guideline-recommended strategies compared with patients without consultations [19]. Although comprehensive ASP with additional ID specialist consultations may be burdensome, it might be necessary for the “optimal” antibiotic use in patients within the last two weeks of their life.

Newer antibiotics, such as ceftazidime-avibactam or ceftolozane-tazobactam, which can be used against carbapenem-resistant Gram-negative bacteria, had not been introduced in South Korea during the study period. Therefore, regimens combining carbapenem with other agents might have been used to treat carbapenem-resistant Gram-negative bacterial infections. One study showed that following the implementation of carbapenem-focused antimicrobial stewardship programmes, there was a decrease in the use of carbapenems but an increase in the use of ceftazidime-avibactam and ceftolozane-tazobactam [20]. Ceftolozane-tazobactam has recently become available in South Korea, but other newer antibiotics are still not available. As carbapenem-resistant Gram-negative bacterial infections continue to increase, newer antibiotics to treat infections caused by these strains must be introduced urgently.

The combination of ID consultation and ASP also appears to enhance appropriate therapy in terminally ill patients [21,22]. In a study involving 459 patients in their terminal stages of illness, it was found that cessation of antibiotics after interventions by ASP did not result in higher mortality rates [23]. The ASP team, which included clinical pharmacists and ID physicians, conducted audits of patients who were prescribed intravenous antibiotics, including carbapenem, fluoroquinolones, and piperacillin-tazobactam [23]. Frequent antibiotic consultant interventions may help decrease carbapenem use even in terminally ill patients.

Our study had a number of limitations. First, the computerised antibiotic control program adherence to treatment recommendations was not objectively investigated. The effect of ASP on the behaviour of professionals varied considerably across studies [24]. Second, physicians at each hospital participating in the evaluation assessed the optimality of antibiotics, which could lead to variations in their evaluations. Nevertheless, we attempted to address this limitation by enlisting the expertise of an ID specialist. Third, this research was unable to establish causality definitively as a retrospective and epidemiological study. Last, detailed information regarding patient comorbidities was not available, and no assessment of the stage of illness or stay in intensive care unit was conducted. The definition of the last days of life was not clear due to patient characteristics; however, all deceased patients were included because the goal of the study was to examine the last antibiotic administered regardless of the patient’s disease status.

## 4. Materials and Methods

### 4.1. Study Setting and Population

The multicentre retrospective cohort study was conducted in 14 South Korean teaching hospitals between 1 November 2018 and 31 December 2018. The study only included patients above 18 years of age who died during the study period at each hospital, identified through a review of their electronic medical records. All centres had appointed infectious disease (ID) specialists. During the study period, carbapenem use was universally controlled by a computerised antibiotic control program. The ID specialists provided recommendations on carbapenem use or discontinuation within 72 h, based on the clinical status and microbiological results, through a chart review. A follow-up evaluation was performed after the approved period of carbapenem use, which was set by the ID specialist. The Institutional Review Board of Kyungpook National University Chilgok Hospital (KNUCH 2019-09-008) and all the participating hospitals granted approval for the study, and the requirement for informed consent was waived given the retrospective observational nature of the study involving deceased patients.

### 4.2. Data Collection and Definitions

The following data were obtained from the patient medical records using a standardised case report form: demographics, date of completion of life-sustaining treatment (LST) documents, class and number of antimicrobial agents used during the last two weeks of life, use of antimicrobial agent on the day of death, ID specialist consultation, microbiological testing, and isolation of MDR organisms. ID specialists retrospectively assessed carbapenem prescriptions without formal ID consultations as needing escalation, de-escalation, continuation, discontinuation, or not assessable in accordance with antibiograms of isolated pathogens, clinical practice guidelines for infectious diseases, and the Sanford guide to antimicrobial therapy. The appropriateness of antibiotic prescriptions was based on the class and did not consider route of administration or dose. Cases assessed as “needing continuation” were considered as having “optimal” antibiotic prescription; others including needing escalation, de-escalation, and discontinuation were classified as no having “not optimal” antibiotic prescriptions. MDR organisms included were methicillin-resistant Staphylococcus aureus (MRSA), vancomycin-resistant enterococci (VRE), MRPA, MRAB, and CRE.

### 4.3. Statistical Analysis

Frequencies and percentages were used to present discrete data, while continuous variables were expressed as mean ± standard deviation or as median and interquartile range following the Shapiro–Wilk normality test. The χ2, Fisher’s exact, two-sample t-, or Mann–Whitney U-tests were used as appropriate to compare characteristics between subgroups of “optimal” and “not optimal” carbapenem prescribing practice. A multivariable logistic regression model was employed to identify predictors of carbapenem use. Variables with a *p* value < 0.05 in the bivariate analysis were included in the subsequent multivariable analysis. The Hosmer–Lemeshow statistic was used to evaluate the goodness of fit of the final model. Factors associated with “not optimal” carbapenem administration were also analysed using a logistic regression analysis. When the distribution of continuous data was skewed, log transformations were applied for univariate analyses. IBM SPSS Statistics for Windows (version 23.0; IBM Corp., Armonk, NY, USA) was used to perform all analyses.

## 5. Conclusions

In conclusion, carbapenems are widely administered to patients in their last days of life, and a considerable proportion of them are administered inappropriately. Frequent consultation with antibiotic specialists in addition to antimicrobial control programs may be necessary to ensure appropriate carbapenem use.

## Figures and Tables

**Figure 1 antibiotics-12-00964-f001:**
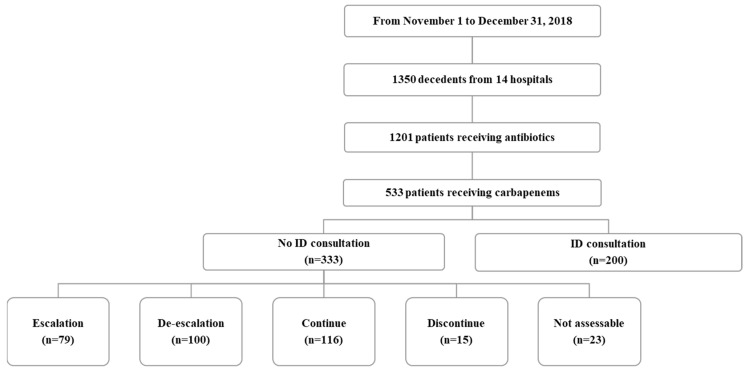
Antimicrobial use in the 14 days before death.

**Table 1 antibiotics-12-00964-t001:** Characteristics of patients receiving carbapenems in the 14 days before death.

	Carbapenem Use (n = 533)	Non-Carbapenem Use (n = 668)	*p* Value
Age, years, median (IQR)	71 (61.0–79.0)	72 (63.0–80.0)	0.158
Gender, n (%)			0.159
Male	338 (63.4)	397 (59.8)	
Female	195 (36.6)	271 (40.8)	
Underlying disease, n (%)			
Cancer	224 (42.0)	303 (45.4)	0.247
Cardiovascular disease	30 (5.6)	61 (9.1)	0.023
Renal disease	17 (3.2)	12 (1.8)	0.118
Chronic lung disease	20 (3.8)	25 (3.7)	0.993
Diabetes	13 (2.4)	9 (1.3)	0.161
Cerebrovascular disease	44 (8.3)	92 (13.8)	0.003
Liver disease	17 (3.2)	27 (4.0)	0.435
Gastrointestinal disorder	14 (2.6)	9 (1.3)	
Cause of death, n (%)			
Any infectious disease	341 (64.0)	193 (30.6)	<0.001
Cancer	110 (20.6)	210 (33.3)	<0.001
Cerebrovascular disease	17 (3.2)	68 (10.8)	<0.001
Cardiovascular disease	20 (3.8)	54 (8.6)	0.001
Lung disease	9 (1.7)	43 (6.8)	<0.001
Liver disease	10 (1.9)	27 (4.3)	0.023
Renal disease	9 (1.7)	20 (3.2)	0.116
Gastrointestinal bleeding	8 (1.5)	16 (2.5)	0.231
The completion of LST document prior to death, n (%)			
LST document completed ≤14 days prior to death	359 (67.4)	465 (69.6)	0.403
LST document completed >14 days prior to death	46 (8.6)	70 (10.5)	0.281
Microbiological study, n (%)	510 (95.7)	556 (83.2)	<0.001
Multidrug-resistant pathogen	205 (40.2)	114 (20.5)	<0.001
Number of antibiotic changes, median (IQR)	4 (3–5)	2 (1–3)	<0.001
ID specialist consultation, n (%)	200 (37.5)	127 (19.0)	<0.001
No ID specialist consultation, n (%)	333 (62.5)	541 (81.0)	<0.001
Escalation	79 (23.7)	74 (13.7)	
De-escalation	100 (30.3)	69 (12.8)	
Continue	116 (34.8)	205 (37.9)	
Stop	15 (2.8)	144 (26.6)	
Not assessable	23 (4.3)	49 (9.1)	

Abbreviations: IQR, interquartile range; LST, life-sustaining treatment; CRE, carbapenem-resistant Enterobacteriaceae; ID, infectious disease.

**Table 2 antibiotics-12-00964-t002:** Comparison of characteristics between patients prescribed “not optimal” and “optimal” carbapenem without ID consultation (n = 310).

Patients’ Characteristics	Not Optimal Use (n = 194)	Optimal Use (n = 116)	Unadjusted OR (95% CI)	*p* Value
Carbapenem treatment duration, days	7 (3–12)	6 (3–12)	1.01 (0.96–1.04)	0.882
Age (years), median (IQR)	74.0 (60.0–79.0)	72.0 (62.8–80.0)	1.01 (0.99–1.02)	0.601
Gender, n (%)				
Male	128 (66.0)	69 (59.5)	1.32 (0.82–2.12)	0.251
Female	66 (34.0)	47 (40.5)	0.76 (0.47–1.22)	0.757
Underlying co-morbidities, n (%)				
Cancer	97 (50.0)	54 (46.6)	1.15 (0.72–1.82)	0.557
Cardiovascular diseases	15 (7.7)	4 (3.4)	2.35 (0.76–7.25)	0.138
Renal diseases	10 (5.2)	2 (1.7)	3.10 (0.67–14.39)	0.149
Lung diseases	8 (4.1)	4 (3.4)	1.20 (0.36–4.09)	0.766
Diabetes	3 (1.5)	2 (1.7)	0.90 (0.15–5.44)	0.904
Cerebrovascular diseases	10 (5.2)	4 (3.4)	1.52 (0.47–4.97)	0.487
Liver diseases	4 (2.1)	5 (4.3)	0.47 (0.12–1.78)	0.264
Gastrointestinal disorders	3 (1.5)	2 (1.7)	0.90 (0.15–5.44)	0.904
Cause of death, n (%) (n = 304)				
Any infectious disease	118 (62.1)	69 (60.5)	1.07 (0.66–1.72)	0.784
Cancer	44 (23.2)	31 (27.2)	0.81 (0.47–1.38)	0.430
Cerebrovascular disease	4 (2.1)	0		0.999
Cardiovascular disease	10 (5.3)	5 (4.4)	1.21 (0.40–3.64)	0.733
Lung disease	3 (1.6)	3 (2.6)	0.59 (0.12–2.99)	0.527
Liver disease	5 (2.6)	2 (1.8)	1.51 (0.29–7.93)	0.624
Renal disease	3 (1.6)	2 (1.8)	0.90 (0.15–5.46)	0.907
Gastrointestinal bleeding	3 (1.6)	2 (1.8)	0.90 (0.15–5.46)	0.907
LST document completion prior to death, n (%)				
Completed ≤14 days prior to death	142 (73.2)	83 (71.6)	1.09 (0.65–1.81)	0.754
Completed >14 days prior to death	15 (7.7)	6 (5.2)	1.54 (0.58–4.08)	0.389
Not completed	37 (19.1)	27 (23.3)	0.78 (0.44–1.36)	0.777
Number of antibiotic changes, median (IQR)	3.0 (2.0–4.0)	3.5 (3.0–5.0)	0.83 (0.71–0.97)	0.023
Microbiological testing, n (%)	183 (94.3)	110 (94.8)	0.91 (0.33–2.52)	0.852
Multidrug-resistant pathogen (n = 293)	57 (31.1)	40 (36.4)	0.79 (0.48–1.30)	0.359

Abbreviations: ID, infectious disease; OR, odds ratio; CI, confidence interval; IQR, interquartile range; LST, life-sustaining treatment.

## Data Availability

Data are available on request due to restrictions.

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
