# Peer review of "Carbapenem Use in the Last Days of Life: A Nationwide Korean Study"

_antibiotics, 2023, doi:10.3390/antibiotics12060964_

Round 1
Reviewer 1 Report
The study reports on a study that investigated the use of carbapenems in patients in the last 14 days of their lives, with the aim of improving protocols for the appropriate carbapenem use. Of those without consultation, 58.2% were assessed as "not optimal" in their use of carbapenems. The study suggests that carbapenems are commonly prescribed to patients in their last days of life, many of whom may not require them, despite existing restrictions. I think this manuscript includes a unique study concept, and I present several comments that might improve this manuscript.
My comments are as follows;
1) Introduction
Based on the purpose of the research presented by the authors, it would be necessary to compare carbapenem vs. non-carbapenem. Furthermore, by analyzing the factors for prescribing carbapenem in those two groups, there is a possibility of identifying specific measures for the future.
2) Table 1
Despite administering carbapenems, the proportion of death due to infection is as high as 64%. The authors should clearly state the reason for this status.
3) Discussion L128-
I understood that the carbapenem prescription has been universally restricted by a computerized antibiotic control program in Korea. Nevertheless, this study showed a high prescription rate and inappropriate use of carbapenems. Please provide the author's insights/discussions on the reasons why this system is not functioning properly.
4) Methods 4.2. L197-201
Please provide further detail on the definition and validity of the "optimal" and "not optimal" in this study. In particular, since it is unclear how "not optimal" is determined, please clarify.
Here is a minor comment
5) Methods 4.3.
Although the multivariate method was mentioned in Methods 4.3., there were no results of the multivariate analysis presented.
Reviewer 2 Report
Very timely and appropraite paper.
Author Response
Dear Sir:
I really appreciate you for helpful comment.
Sincerely, Ki Tae Kwon
Reviewer 3 Report
Talking about the use of carbapenems is a very, very used topic, so it may not be particularly interesting. However, I think it's well phrased and brings a bit of new information about it.
In any case, I miss a more detailed study with a much larger sample, because I think it is feasible through collaborations.
Congratulations for taking it to ECCMID.
Author Response

(The authors gave the same response as above.)

Reviewer 4 Report
In a post-hoc analysis of a national database, Wi et al. found that carbapenems were frequently used in the last 14 days of life of deceased patients despite an AB control programme. Two-thirds of prescriptions are without ID consultation and two-thirds of prescriptions without ID consultation are not fully appropriate.
This study suggests that an AB control programme is not effective. An antibiotic stewardship programme with or without ID consultation may be more effective.
Major: the justification for a specific analysis during the last day of life is unclear to me? it leads to an obvious historical bias, since death is not planned when carpapenem is prescribed. please add a justification? it would have been preferable to use a subgroup of very severe patients. the rate of microbiological sampling before treatment is high.
The rate of pre-treatment microbiological sampling is very high. This argues for appropriate use of antimicrobials.
ICU stay: The rate of inappropriate use of antibiotics is considered to be lower in the ICU compared to the ward. What was the effect of ICU stay on the rate of appropriate carbapenem use?
The cause of death was an infectious disease in about two-thirds of cases, what was the rate of inappropriate carbapenem use in this specific subgroup?
The reason for the lack of appropriate escalation or de-escalation should be detailed. The definition of inappropriate escalation, de-escalation should be added in detail. The dose of carbapenem may also be suboptimal. Does the ID assess retrospectively whether the dose was appropriate?
in the discussion, please add a section on the absence of positive impact of prescription control and explain why it failed.
title: please change: i may suggest: "Carbapenem use in the last days of life: a nationwide Korean study"
Round 2
Reviewer 1 Report
Thank you for the response to the comments.
I have no further comment.